# Impact of Climate Change on the Energy Consumption of Passenger Car Vehicles

**Samuel Hasselwander** [1,*] **, Anton Galich** [2] **and Simon Nieland** [2]

1   Institute of Vehicle Concepts, German Aerospace Center (DLR), Wankelstraße 5, 70563 Stuttgart, Germany
2   Institute of Transport Research, German Aerospace Center (DLR), Rudower Chaussee 7,
    12489 Berlin, Germany; anton.galich@dlr.de (A.G.); simon.nieland@dlr.de (S.N.)
*   Correspondence: samuel.hasselwander@dlr.de; Tel.: +49-711-6862-8469

**Abstract:** The energy consumption of passenger vehicles is affected by the physical properties of the environment. The ambient temperature in particular has a significant impact on the operating energy consumption. To quantify the impact of a changed climate on vehicles with different drivetrain systems, we set up a model that calculates the change in energy demand with respect to multiple global warming levels. In particular, the effect of rising temperatures on the energy consumption of battery electric vehicles and vehicles with internal combustion engines was investigated. Our results indicate that climate change will likely lead to a rise in energy consumption of vehicles with an internal combustion engine. This is mostly due to the increase in cabin climatization needs caused by the higher ambient temperatures. At a global warming level (GWL) of 4.0 °C, the calculated annual energy consumption on average is 2.1% higher than without taking the climate-change-related changes in temperature into account. Battery electric vehicles, on the other hand, are expected to have a lower overall energy consumption (up to −2.4% at 4 °C GWL) in cold and moderate climate zones. They benefit from the lower heating needs during winter caused by global warming.

**Keywords:** climate change; vehicles; passenger car; battery electric; energy consumption; impact

## 1. Introduction

Analyzing the impact of different factors on the energy consumption of passenger vehicles with different drivetrains constitutes an important issue in the research community. For instance, many have focused on the differences in laboratory versus real-world energy consumption [1–3]. The ambient temperature constitutes another important factor influencing the operating energy consumption of the vehicle and the various drivetrains.

Different temperatures at the cold start of the drive system or different air conditioning requirements while driving can affect the energy consumption of a vehicle significantly [4]. The driving range of battery electric vehicles, for instance, can decrease by up to 31.9% due to heating and by up to 21.7% due to limited recuperation under cold conditions [5]. Plug-in hybrid electric vehicles can suffer from shorter driving ranges if operated in cold ambient temperatures [6]. Hence, ambient temperature can have a considerable impact on the energy consumption of passenger vehicles, and this impact can vary in different temperature conditions.

Against this background, it is evident that climate change will significantly affect the energy consumption of passenger vehicles on the basis of altered weather conditions. Among others, it is expected that the global surface temperature will continue to rise and that global warming will exceed 2 °C in the 21st century under all emission scenarios due to climate change [7]. Under the scenarios with excess greenhouse gas emissions, the increase in global surface temperature over the 20 year period between 2081 and 2100 is projected to be between 3.3 and 5.7 °C [7].

In Germany, the amount of frost days is likely to fall between 21–32% in the next 40 years, while the amount of summer days is likely to increase by up to 53% [8]. It

therefore can be assumed that there will be less vehicle cold starts and more cabin climatization needs in the future, which will both significantly affect the energy consumption of passenger vehicles.

However, the impact that different climate change scenarios might have on the energy consumption of passenger vehicles is poorly researched. In fact, only one study that directly addresses this issue could be found. Based on panel data of consumer expenditure on gasoline and actual gasoline sales, ref. [9] estimate that fuel consumption in the USA might increase between 1.63% and 3.95% under different climate change scenarios by the end of the 21st century. Unfortunately, the study does distinguish between different drivetrain technologies and fuels but only considers gasoline consumption.

This study attempts to contribute to filling this gap in the literature by providing insights on the potential impact of climate change under different emission scenarios on the energy consumption of passenger vehicles with different drivetrain technologies. For this purpose, we set up a model to calculate the change in energy consumption based on exemplary mobility data in different climate zones worldwide as well as on actual German mobility data with respect to the scenarios of global warming (1.5, 2.0, 2.0 and 4.0° Celsius), defined by the Intergovernmental Panel on Climate Change (IPCC). Other factors that affect the real world vehicle energy consumption, such as extra mass, real world aerodynamics and additional consumers excluding the heating ventilation air conditioning (HVAC) unit, etc. as listed in [1] are not expected to change due to effects of climate change and therefore are not considered in this study.

By investigating the impact of climate change on the energy consumption of passenger vehicles with different drivetrain technologies, this paper also addresses a concrete real-world problem, as the transport sector contributes almost a quarter of Europe's greenhouse gas emissions [10]. If climate change, for instance, is likely to increase the energy consumption of certain drivetrain technologies, while decreasing the energy consumption and thus the greenhouse gas emissions of others, then policy-makers could be recommended to introduce measures that promote the uptake of vehicles with drivetrain technologies that are likely to have a less negative ecological impact.

This brief introduction is followed by a section that describes the different sources of data underlying the analysis in this paper. Subsequently, Sections 3 and 4 outline the methods and the results of the study at hand. As this is one of the first studies investigating the impact of climate change on the energy consumption of passenger vehicles, not all potentially interesting factors could be addressed. Therefore, the limitations of the proposed approach are discussed in Section 5, which simultaneously illustrates the need for further research. Finally, Section 6 summarizes the main conclusions of this paper and the practical relevance.

## 2. Data

First, the mobility, weather and climate data underlying this study are briefly introduced. In particular, we describe how a mobility data set was enriched with information on local weather conditions and outputs of regional climate models. Thereafter, the basic mechanisms of technologies generally used in the two main processes for regulating the cabin temperature of passenger cars, i.e., cooling and heating, are described in more detail. Finally, we explain how the additional energy consumption caused by cold starts with different drivetrain technologies is considered in this study.

### 2.1. MiD Data Set Combined with Regional Climate Models to Generate Future Weather Scenarios

The B3 local data set of the survey "Mobility in Germany 2017" (MiD) constitutes the most important source of data used in the project [11,12]. The main objective of the survey was to capture people's mobility behaviour on an average day in Germany. For this purpose, a specific sample design was developed so that the survey participants are representative of the population of Germany in terms of age and sex as well as federal state and spatial area of the place of residence.

To correct minor over- or underrepresentation, the data set contains specific person and trip weights. Furthermore, the participants were assigned different days of the week in different months and seasons of the year to complete their travel diaries. Hence, travel diaries were reported for different days from May 2016 to September 2017 so that daily, weekly, monthly and seasonal effects could be included evenly in the data set.

Altogether, around 316,000 people from 156,000 households contributed their travel information, including trip origins and destinations that were spatially located in the inspire grid system of the European Union [13]. Yet, not all of the survey participants reported their trip origins and destinations, and thus the data set does not include grid cells for all of the trips reported.

However, in particular the grid cells of the points of departure were needed to identify the nearest weather stations so that the weather conditions prevailing at the time and location of the trip starts could be accurately added to the data set. Against this background, we decided, for trips not longer than 5 km, that the grid cell of the point of arrival can be used if the grid cell of the point of departure is missing in order to have more trips in the final data set. The distance of 5 km was deemed as short enough so that the weather conditions at the point of arrival of a trip should be the same or very similar to those at its point of departure.

Consequently, all trips that did not provide information on the grid cell of the point of departure and that were not longer than 5 km and did not provide information on the grid cell of the point of arrival were excluded from the data set. The spatial distribution of the points of departures of the remaining trips is illustrated in Figure 1:

The remaining 533,906 trips were made by 214,558 people. Unfortunately, the exclusion of many trips brought about a bias in the final data set in terms of trip length, day of the week on which the trip was conducted, the sex and age of the participants and further variables.

Therefore, an iterative proportional fitting procedure was developed to adjust the person and trips weights so that they meet the actual marginal distributions of the variables sex, age, driving license, car availability, day of the week of the trip start, month of the trip start, trip start time, trip arrival time, trip purpose, trip duration, trip length, household size, number of cars in the household, economic status of the household, spatial area type and mode of transport in the original data set. The adjusted person and trip weights ensure that the final data set after the exclusion of many trips still represents people's mobility behaviour on an average day in Germany.

Based on the grid cells and the date and time of the trip starts, the local air temperature measured two metres above ground at the nearest weather station of the German Weather Service was added to the data set. In addition, the air temperature expected to be in place at the locations of the trip starts and on the dates of these if the global warming levels (GWL) of 1.5, 2.0, 3.0 and 4.0 °C are reached were calculated on the basis of the outputs of regional climate models. The regional models were developed and calculated by the EURO-CORDEX initiative, which aims at downscaling global climate projections to a regional scale (12 km ground resolution) for the European continent [14]. The details of this procedure can be read in [15].

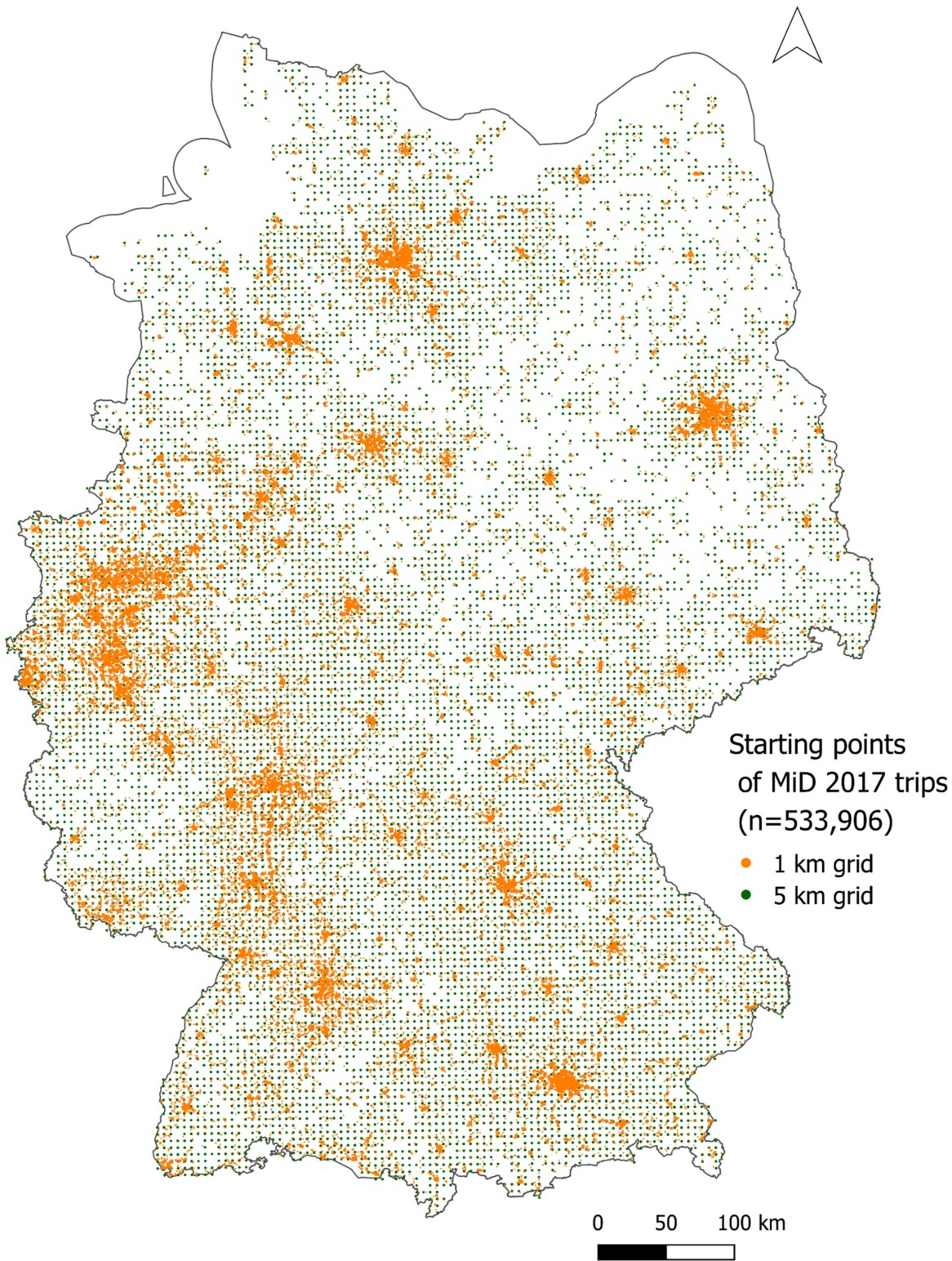

**Figure 1.** The distribution of the points of the trip departures in Germany is shown. The green dots indicate the centroid of the trip departures, which were available in a 5 km grid cell, while the orange dots denote the centroid of the trip departures, which were available in 1 km grid cells. The size of the grid cells was determined by privacy protection regulations, i.e., if only few people lived in a 1 km grid cell containing the point of departure of a certain trip, then for that trip the respective 5 km grid cell of the inspire grid system was chosen in order to have more residents within the grid cell and to make the identification of the actual person that conducted the trip in question more difficult.

The described steps of data prepation resulted in a data set of trips conducted with passenger vehicles that includes the air temperature measured at the point and time of departures as well as the air temperature expected to be in place at that point on that day of the year if the global warming levels of 1.5, 2.0, 3.0 and 4.0 °C are reached. Table 1 illustrates an excerpt of the resulting data set:

**Table 1.** Excerpt from the trip data set.

| Person ID | Trip ID | Grid Cell ID of the Trip Start | Trip Length (in km) | Date and Time of the Trip Start | Air Temperature Measured at Trip Start (in °C) | Air Temperature Expected at Trip Start in GWL 1.5 (in °C) | Air Temperature Expected at Trip Start in GWL 2.0 (in °C) | Air Temperature Expected at Trip Start in GWL 3.0 (in °C) | Air Temperature Expected at Trip Start in GWL 4.0 (in °C) |
|---|---|---|---|---|---|---|---|---|---|
| 1 | 1 | 5 km N2885E4365 | 6.44 | 9 November 2016 10:30 | 2.9 | 4.5 | 4.5 | 5.4 | 6.5 |
| 1 | 2 | 5 km N2885E4365 | 6.44 | 9 November 2016 11:45 | 3.2 | 4.8 | 4.8 | 5.7 | 6.8 |
| 2 | 1 | 1 km N2939E4167 | 18.05 | 7 July 2017 06:15 | 17.7 | 19.2 | 19.6 | 21.6 | 22.3 |
| 2 | 2 | 500 m N29270E41590 | 18.05 | 7 July 2017 15:20 | 31.0 | 32.5 | 32.9 | 34.9 | 35.6 |
| 2 | 3 | 1 km N2939E4167 | 0.95 | 7 July 2017 16:10 | 29.4 | 30.9 | 31.3 | 33.3 | 34.0 |
| 3 | 1 | 500 m N30810E40450 | 2.7 | 24 January 2017 08:30 | −0.9 | 0.1 | 0.2 | 0.9 | 2.1 |
| 3 | 2 | 500 m N30810E40450 | 2.7 | 24 January 2017 16:30 | 0.1 | 1.3 | 1.4 | 1.9 | 2.9 |

*2.2. Fuel Consumption of Heating Ventilation Air Conditioning Unit*

A heating ventilation air conditioning (HVAC) unit contains a fan, a heating and a cooling heat exchanger. For vehicles with an internal combustion engine (ICE), the heating heat exchanger is usually fed by the hot cooling fluid of the engine and therefore using the waste heat of the combustion process to heat the cabin intake airflow. Battery electric vehicles (BEVs) on the other hand have to heat up the cabin either with a positive temperature coefficient (PTC) heater or a heat pump (HP) because of their lack of a heat source, such as the ICE. PTC-heaters are a cheap, simple and fast solution for heating up the cabin intake airflow and therefore already featured in battery electric vehicles with performance maps displayed in the scientific literature as shown in [16,17].

Heat pumps on the other hand are more complex and expensive compared to PTC-heaters but also more efficient, which results in higher winter driving ranges for BEVs. A HP can have multiple configurations. Regarding the heat source, for example, it can use the battery, the electric motor, the inverter or the ambient air or even all of these sources combined, which results in various different performance maps. Despite more manufacturers offering BEVs with a HP option, the scholarly literature featuring performance maps of passenger car heat pumps is rather thin. Li et al. showed the advantages of a HP over an PTC-heater pointing out that the driving range at −10 °C was 23% higher with the use of a HP system [18].

The cooling heat exchanger typically works as an air conditioning (AC) device in conventional vehicles. This consists of a closed circuit containing a compressor, a condenser, an expansion valve and an evaporator with a refrigerant circulating. The compressor is driving the system powered externally either by a belt driven directly from the ICE or via an electric motor in case of an electric vehicle. Compressed by the compressor the refrigerant condensates isobar in the condenser and cools down after the expansion at the expansion valve. In the evaporator, the refrigerant absorbs the heat flow of the intake air while evaporating and therefore cools down the cabin intake airflow.

Since there are many different technical possibilities for heating and cooling heat exchangers of a HVAC, with different heat sources and performance maps, we chose a more generic approach for how to measure the effects of a rising ambient temperature on passenger car vehicles. For calculating the cabin thermal energy needs, we used transient and steady-state power numbers based on a cabin thermal model developed by Mansour

et al. [19] for a mid-sized passenger car. In this case, the transitional condition refers to the initial cooling or heating after starting the vehicle when it is soaked at low or high temperatures.

The steady-state is reached after the target cabin temperature of 23 °C has been reached. The thermal power for this operation is the sum of the power of the AC compressor, the heater and the fan. The air mass flow rate of the fan depends on the ambient temperature and can be taken from Mansour et al. [19] (180–408 kg·h$^{-1}$ in steady state and 285–640 kg·h$^{-1}$ in transient state. The recirculation ratio is set to zero in heating mode and increases with the ambient temperature in cooling mode (from 50% at 25 °C to 88% at 40 °C) [19]. The power numbers for reaching a target cabin temperature of 23 °C at different ambient temperatures are displayed in Figure 2 (red lines for heating power and blue lines for cooling power needs).

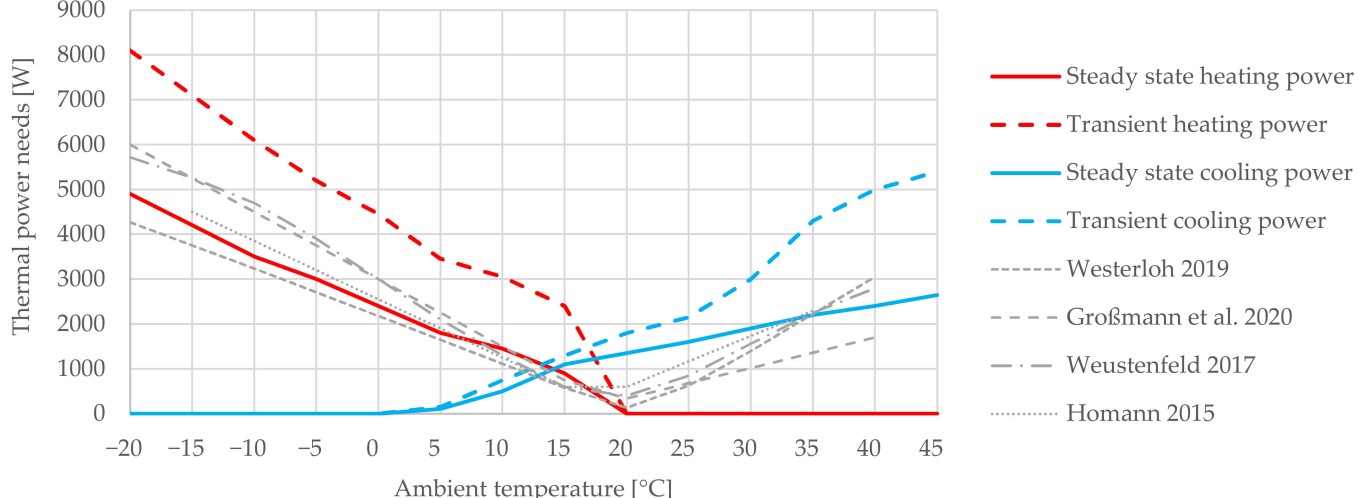

**Figure 2.** Used heating power needs are shown in red lines and used cooling power needs in blue lines for a mid-sized passenger car based on a cabin thermal model developed by Mansour et al. [19]. They are displayed in comparison to other studies with steady state power numbers from Westerloh [16], Großmann et al. [17], Weustenfeld [20] and Homann [21], which are shown in grey lines.

This is furthermore compared to several other, occasionally less extensive studies from Westerloh [16], Großmann et al. [17], Weustenfeld [20] and Homann [21] with partial different assumptions in cabin temperature, vehicle size, number of passengers and solar irradiation. The model of Mansour et al. [19] was chosen because of the availability of a detailed power map over the ambient temperature also featuring transient power needs while still being comparable to existing literature. The assumptions of Mansour et al. are relatively conservative:

They calculate without passengers and without taking solar radiation into account in heating mode but with four passengers and a solar radiation of 700 W/m$^2$ in cooling mode. However, if we analyze the energy demand as a function of ambient temperature, it corresponds to other data from the scientific literature (see Figure 2). Since there is less scientific literature especially for the transient demand, we decided to use the generic model of Mansour at al. and thus also adopted their assumptions.

The convergence time between transient and steady state mode was determined by experimental measurement and ranges between maximal 20 min for initial temperatures of −15 °C or lower and 2.5 min at an initial temperature of 20 °C. At 45 °C Mansour et al. measured a convergence time of 15 min. In between those reference temperatures, the other convergence times have been interpolated linearly [19].

Currently, battery electric vehicles are increasingly being fitted with heat pumps. BEVs equipped with heat pumps typically still have a PTC for a fast warm up of the cabin (transient operation) [21]. Therefore, a sensitivity analysis considering a heat pump only

for the steady state heating of the battery electric vehicle (based on Homann [21]) well as an analysis regarding a full-scale air conditioning heat pump system for both steady and transient heating operation was conducted (based on Li et al. [18]).

Homann [21] describes the experimental results of an air and water to air heat pump, which uses both the ambient air temperature as well as the cooling circuit of the electric drivetrain components. Li et al. [18] demonstrated the use of a full-scale experimental air conditioning heat pump system consisting of three heat exchangers, an air to air heat pump and an additional PTC heater in a small electric vehicle. The different heating power needs for these systems are described in Figure 3.

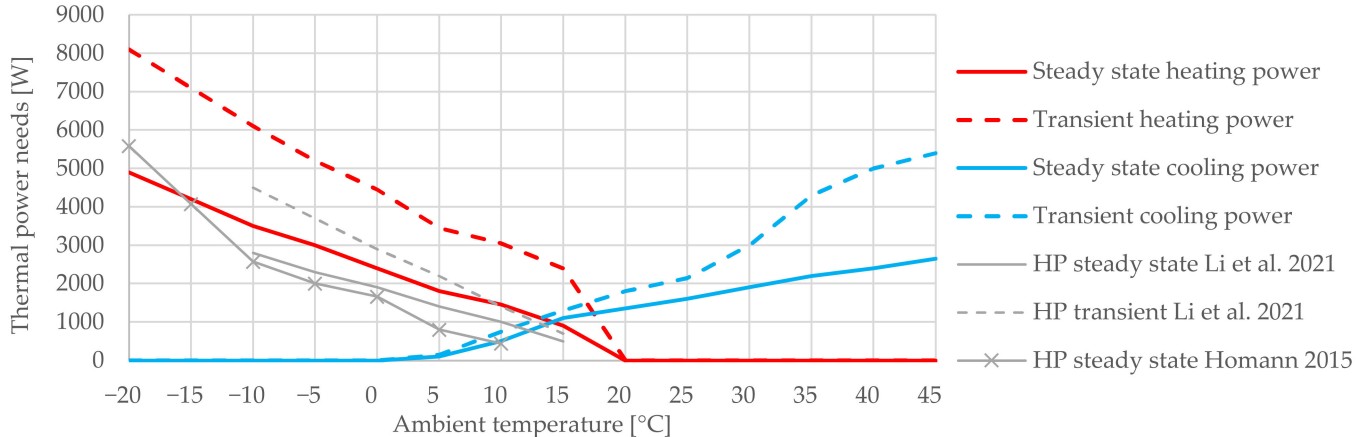

**Figure 3.** Heating power needs are shown in red lines and cooling power needs in blue lines for a mid-sized passenger car based on a cabin thermal model developed by Mansour et al. [19] in comparison to other studies that featured battery electric vehicles with heat pumps (HP) from Li et al. [18] and Homann [21].

It is noticeable that the power needs for the HP-based systems are slightly below the used heating power needs based on Mansour et al. [19]. This is mainly because of the use of the more efficient heat pumps but can also be caused by the focus on different vehicles and slightly different climatization conditions of the considered literature. Therefore, the HP related evaluations can only be taken as a sensitivity analysis to the main evaluation.

### 2.3. Additional Fuel Consumption for Different Drivetrain Systems Caused by Cold Start Events

The ambient temperature determines the starting temperature of a vehicle after a sustained parking period. A cold start usually takes place when the temperature of the vehicle components at vehicle start is below their operation temperature. During the warm up phase, the powertrain components, such as the ICE, the transmission, the battery, the electric motor and the lubricants operate differently, which results in increased energy consumption [4].

The amount of consumption deviation depends strongly on the starting temperature and the trip length. In particular, short-distance trips after a cold start, when the operating temperature cannot be reached, tend to have a higher energy consumption than comparable trips that were made with a warmed-up powertrain [22]. The duration of the warm up phase strongly depends on the specific vehicle and the different powertrain components. Battery electric vehicles have lower operation temperatures than vehicles with an internal combustion engine.

While the battery of a BEV works most efficient in a temperature range of 15–35 °C [23], the operation temperature of an internal combustion engine is at around 90 °C [24]. Therefore, and because of the different energy conversation and system efficiencies of these powertrains (e.g., burning fuel vs. direct use of electric energy), they tend to have different consumption variances related to the starting temperature [25]. We therefore decided to

derive different trendlines regarding the change in fuel consumption of ICEVs and BEVs caused by the vehicle cold start.

To calculate the fuel consumption variance for passenger car vehicles with internal combustion engine we summarized various vehicle testbench data from current literature and DLR-internal measurements. Suarez-Bertoa et al. [26] analized the emission factors of five gasoline and five diesel vehicles according to the 23 km Worldwide harmonized Light vehicles Test Procedure (WLTP) at −7 °C and compared them with emissions at 23 °C. They found that, on average, gasoline vehicles had 9% higher and the diesel vehicles had 15% higher $CO_2$ emissions at −7 °C. Given that the $CO_2$ emissions correlate directly with the fuel consumption, it can be said that fuel consumption at −7 °C is also 9% or 15% higher, respectively.

Zhu et al. [27] also tested two gasoline vehicles at −7 °C in a climate chamber over the WLTP and found a 12% increase in fuel consuption caused by the vehicle starting at this lower operating condition. Bielaczyc et al. [13] analyzed a pool of gasoline and diesel vehicles and ran a full modal emissions analysis of one gasoline and one diesel vehicle over the NEDC. They found a 19% increase in fuel consumption for the gasoline and a 12.5% increase for the diesel vehicle caused by the cold start at −7 °C.

We also used DLR internal vehicle test data out of the EnviTrans project to obtain additional temperature support points. These test bench measurements featured various gasoline and diesel vehicles over the WLTP at different temperatures in the climate chamber. The DLR-internal test cycles as well as the other test cycles ran without cabin heating, as they were conducted under the standard WLTP test cycle regulations.

Apart from conventional vehicles, there is also a collection of fuel consumption data of full hybrid, plug-in hybrid and fuel-cell electric vehicles available, which could be used to further develop additional energy consumption tool in the future. Based on those data points, we calculated an exponential function correlative to Fontaras et al. [1]. The function focusses on the change in fuel consumption of ICEVs compared to the vehicles WLTP norm consumption depending on the starting temperature as displayed in Figure 4.

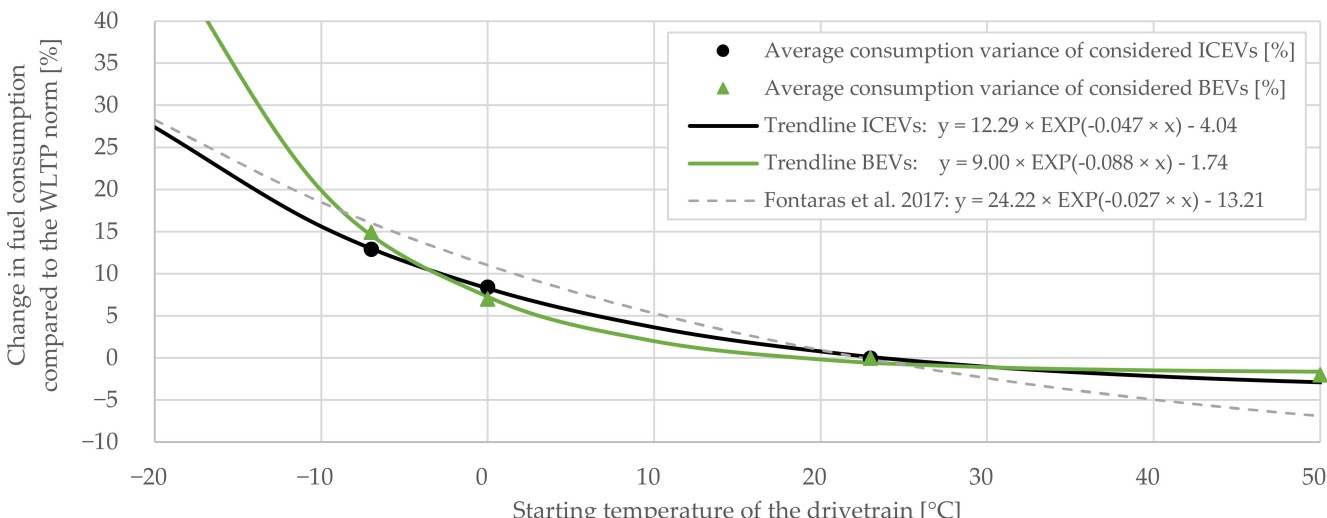

**Figure 4.** Percentage increase in fuel consumption caused by cold starts at different starting temperatures of the drivetrain in the WLTP test cycle compared to a norm start at 22 °C. The trendline function was calculated following Fontaras et al. [1] from four average consumption support points, each being based on several data points from the scientific literature described in Section 2.3 and DLR internal vehicle test bench data.

Regarding the cold start energy consumption of battery electric vehicles, there is less scholarly literature available. The available literature mostly focuses on the additional energy consumption of the cabin heating and different other energy consumers rather than

on the consumption effect of the cold drivetrain itself. In the absence of suitable literature, we extended our search to different driving cycles. The results of Chlopek, Lasocki et al. [28] showed that the FTP-72 driving cycle (Federal Test Procedure) is comparable to the NEDC (New European Drive Cycle) regarding the distance-specific energy consumption.

Therefore, we decided to also work with the results of Meyer et al. [29] who analyzed three different BEVs with and without cabin heating on an vehicle test bench over the FTP-72 and other driving cycles at different temperatures. They found an average increase in energy consumption of 18% at $-7\,°C$ compared to the runs they did at $20\,°C$. This indicates that BEVs are more effected by the vehicle cold start than ICEVs if focusing on energy consumption.

Based on experimental battery cell characteristics and a simulated vehicle model, Liu et al. [30] evaluated the ambient temperature effect on the range of a BEV. They found that the range at $0\,°C$ dropped 7%, at $10\,°C$ dropped $-4\%$ and at $20\,°C$ dropped $-1\%$. With the help of those additional temperature support points, we calculated an exponential function correlative to Fontaras et al. [1] but solely focusing on BEVs. The formula describes the change in fuel consumption compared to the vehicles WLTP norm consumption depending on the starting temperature as displayed in Figure 4.

The average consumption variance caused by a cold start is calculated with a self-developed exponential function following Fontaras et al. [1] for BEVs and ICEVs as described above. Based on the described literature and DLR-internal vehicle testbench data, we first calculated four average support points as displayed as dots and triangles in Figure 4 and thus combined several data points. Afterwards, we fitted an exponential function to those points to obtain a trendline for both ICEVs and BEVs. We also compared the fitted curves to the literature. Our derived trendlines show only slight deviations compared to Fontaras et al. [1], in particular, regarding cold start conditions.

## 3. Methods

To analyze how climate change will affect energy consumption of passenger vehicles a tool was developed to calculate the additional energy consumption of a trip caused by the cabin climatization and the vehicle cold starts. Based on the trip length and actual temperature it calculates the deviation that would be caused if the same trip would take place with climate-change-related changes in temperature.

### 3.1. Energy Need for Climatization

The heating or cooling power requirement is calculated for each trip travelled in the car out of the modified MiD data set on the basis of the characteristic curves from Figure 2. Depending on the ambient air temperature $T_{amb}$ at the beginning of the trip, the required transient and steady state air conditioning power $P(T_{amb})$ is first interpolated. For the calculation of the ambient air temperature at the beginning of the different trips for the different GWLs, we assumed that the temperature increases homogeneously during the day.

Depending on the differential temperature of the corresponding GWL. The transient and steady state climatization durations ere then calculated based on the duration of the different trips. Since it takes different amounts of time for the vehicle to heat up depending on the outside temperature, following [19], the transient time duration $t_{trip,transient}(T_{amb})$ also depends on the ambient air temperature as shown in Equation (1):

$$t_{trip,transient}(T_{amb}) = \begin{cases} 20\ min, & T_{amb} \leq -15\,°C \\ -0.5 \times T_{amb} + 12.5\ min, & -15\,°C < T_{amb} \leq 20\,°C \\ 0.5 \times T_{amb} - 7.5\ min, & T_{amb} > 20\,°C \end{cases} \tag{1}$$

Based on the transient and the overall trip duration the steady state time duration $t_{trip,steady}$ is calculated as shown in Equation (2):

$$t_{trip,\,steady} = \begin{cases} 0, \; t_{trip,transient} \geq t_{trip} \\ t_{trip} - t_{trip,transient}, \; t_{trip,transient} < t_{trip} \end{cases} \tag{2}$$

The energy required for the air conditioning of either a battery electric or an internal combustion engine vehicle is calculated by summarizing the transient and steady state parts of the heating and cooling climatization modes of the different vehicle types $i$ as shown in Equation (3):

$$E_{climatization,i} = E_{trans,heating,i} + E_{steady,heating,i} + E_{trans,cooling,i} + E_{steady,cooling,i} \tag{3}$$

While a BEV will use its battery as an energy source for both heating and for cooling purposes as described in Section 2.2, vehicles with an internal combustion engine are using the waste heat of the ICE for heating with the need for an additional cooling device, such as an air conditioning climate compressor.

This climate compressor is typically belt driven directly from the engine or powered by an electric motor depending on the vehicle concept. Running the AC in a conventional vehicle you will have additional fuel consumption based on the efficiency of the engine and the cooling demands of the HVAC [25]. The energy sources as well as the calculation of the climatization energy for the different drivetrains, climatization modes and dynamics are displayed in Table 2.

**Table 2.** Calculation of climatization energy for different drivetrains, climatization modes and dynamics.

| Vehicle | Climatization Mode | Dynamic | Energy Source | Equation |
|---|---|---|---|---|
| BEV | Heating | Transient | Electric energy out of battery | $E_{trans,heating,BEV} = t_{trip,trans} \times P_{heating,\,trans}(T_{amb})$ |
| | | Steady state | Electric energy out of battery | $E_{steady,heating,BEV} = t_{trip,steady} \times P_{heating,steady}(T_{amb})$ |
| | Cooling | Transient | Electric energy out of battery | $E_{trans,cooling,BEV} = t_{trip,trans} \times P_{cooling,\,trans}(T_{amb})$ |
| | | Steady state | Electric energy out of battery | $E_{steady,cooling,BEV} = t_{trip,steady} \times P_{cooling,steady}(T_{amb})$ |
| ICEV | Heating | Transient | Waste heat of ICE + Electric energy for fan | $E_{trans,heating,ICEV} = t_{trip,trans} \times P_{fan}$ |
| | | Steady state | Waste heat of ICE + Electric energy for fan | $E_{steady,heating,ICEV} = t_{trip,steady} \times P_{fan}$ |
| | Cooling | Transient | Fuel for powering the AC compressor | $E_{trans,cooling,ICEV} = E_{trans,cooling,BEV}$ |
| | | Steady state | Fuel for powering the AC compressor | $E_{steady,cooling,ICEV} = E_{steady,cooling,BEV}$ |

For battery electric vehicles, the heating and cooling energy needs result out of the product of trip duration $t_{trip,j}$ and linearly interpolated climatization power needs $P_{j,k}$ depending on the outside temperature $T_{amb}$ derived out of Figure 2 with $j$ for the different climatization dynamics and $k$ for the different climatization modes. It should be noted that at outside temperatures of 5 to 20 °C, the air conditioner simultaneously heats and cools the cabin intake air to dehumidify the air.

Since the heating energy in ICEVs is the waste heat from the combustion engine, no additional heating power is required. Only the fan for heating up the cabin air needs to be powered. Based on Großmann et al. [17], the electric power requirement of the fan was set at 100 watts for the entire travel distance of all trips with outside temperatures below 20 °C. We assumed that the cooling power requirement is the same as for BEVs calculated on the basis of the blue steady-state and transient power curves in Figure 2. The power demand of the fan to cool the cabin is included in the power demand of the air conditioning as shown in Figure 2. It increases with respect to higher temperatures.

To calculate the additional fuel consumption of the ICEVs climatization needs, an engine efficiency of $\eta_{ICE} = 33\%$ was assumed [31]. Based on the heating value $LHV_{Gasoline}$ and the density $\rho_{Gasoline}$ of the fuel the additional fuel consumption in liters of gasoline is calculated as shown in Equation (4) with $j$ for the different climatization dynamics and $k$ for the different climatization modes:

$$E_{climatization,ICEV} = 1000 \times \sum_{j,k} E_{j,k,ICEV} \times \frac{1}{LHV_{Gasoline} \times \rho_{Gasoline} \times \eta_{ICE}} \ [l] \qquad (4)$$

Based on the $CO_2$ conversion factor $x_{CO_2} = 2.357 \ \text{kg}_{CO_2} \text{l}_{Gasoline}^{-1}$ from the U.S. Environmental Protection Agency [32], we then calculated the additional $CO_2$ emissions created by ICEVs due to the higher climatization needs caused by the climate change.

### 3.2. Energy Needs for Vehicle Cold Start

Based on the exponential trendline displayed in Figure 4, the change in fuel consumption compared to the WLTP norm is calculated depending on the ambient temperature. Based on the norm fuel consumption of the vehicles, as well as the WLTP length of 23.262 km (as shown in Appendix A, Table A1), the absolute additional fuel consumption caused by the cold start is calculated. Assuming that each trip, regardless of its length, has this additional absolute cold start energy consumption as a function of temperature, the relative additional energy consumption caused by the cold start of the respective trip is then calculated as a function of the length of the trip.

### 3.3. Overall Additional Energy Needs

The total fuel consumption results from the sum of the WLTP norm fuel consumption, the relative climatization energy needs as well as the relative additional energy consumption caused by the cold start for each respective trip.

## 4. Impact of Climate Change on the Energy Consumption of Passenger Car Trips in Germany

Assuming all the trips by vehicles with internal combustion engine vehicles out of the MID data with projected fleet distribution of 2040 were made with a VW Golf 1.0l eTSI and all corresponding trips by vehicles with an all-electric powertrain were made with a VW eGolf.

Figure 5 shows the average additional energy consumption of all the trips during the entire year in relation to the vehicle cold-start and HVAC as well as the change in energy consumption if all trips had taken place at a global warming level of 4.0 °C. All in all, the effect of cold-start on the energy consumption of ICEVs is around 5%, which is significantly less than that the almost 13% of vehicle air conditioning. This effect is even greater for BEVs. Here, the additional consumption due to cold starting is nearly 3% compared to an additional consumption of around 27% for the HVAC.

In terms of vehicle cold-starts, both conventional and electric vehicles are expected to benefit from global warming, consuming about 1% less energy. This can be seen from the red bar in Figure 5. The situation is different with regard to the HVAC. Here, the BEV benefits from the fact that less heating power is required due to global warming. Since more energy is required for heating than for cooling, the overall annual energy consumption for BEVs in Germany falls by more than 1% at a GWL of 4 °C. ICEVs, on the other hand, only have the higher air conditioning demand and therefore suffer from the higher energy consumption of the air conditioning system, which could result in more than 3% higher fuel consumption.

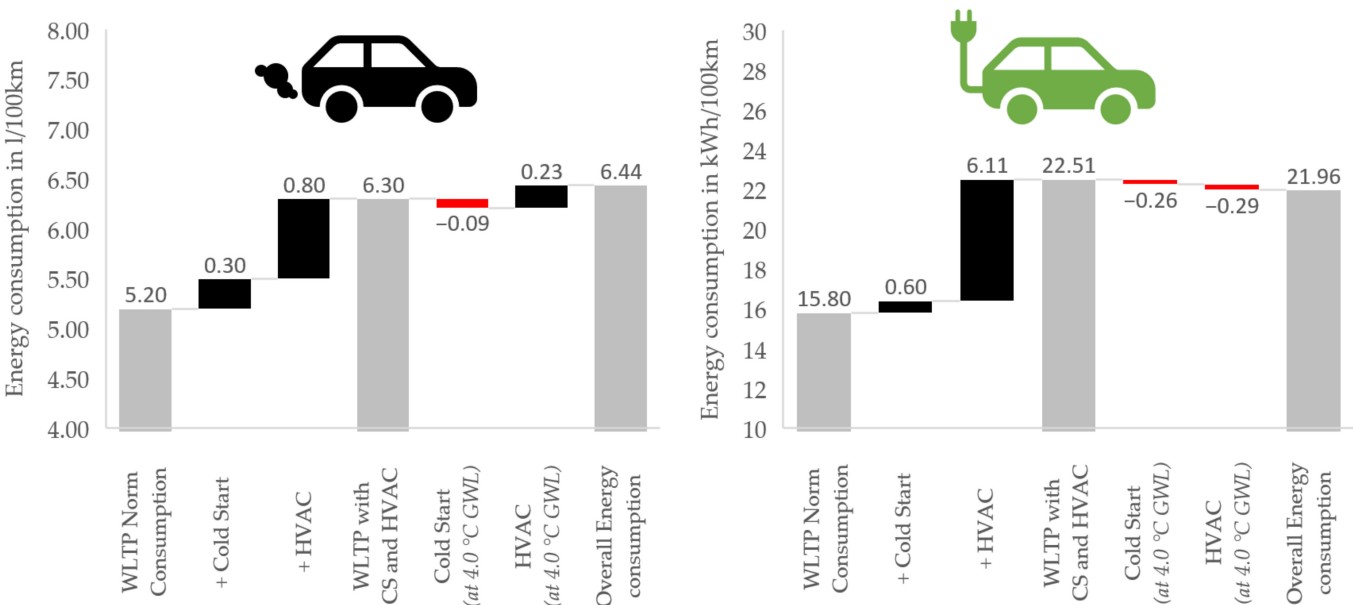

**Figure 5.** Average annual energy consumption of a VW Golf 1.0l eTSI (**left**) and a VW eGolf (**right**) compared to the average calculated energy consumption at 4.0 °C global warming level (GWL) based on the MiD Dataset.

Taking all the trips by vehicles with internal combustion engine vehicles out of the MID data with projected fleet distribution of 2040 into account, we see an additional energy consumption for all global warming levels. This is mainly because of the increase in cabin climatization caused by the higher ambient temperatures. As shown in Figure 6, at a GWL of +4.0 °C, the calculated energy consumption of an ICEV on average is 2.1% higher than without taking the climate-change-related changes in temperature into account. With the current fleet, this would lead to nearly 2 million tons of $CO_2$ released additionally per year.

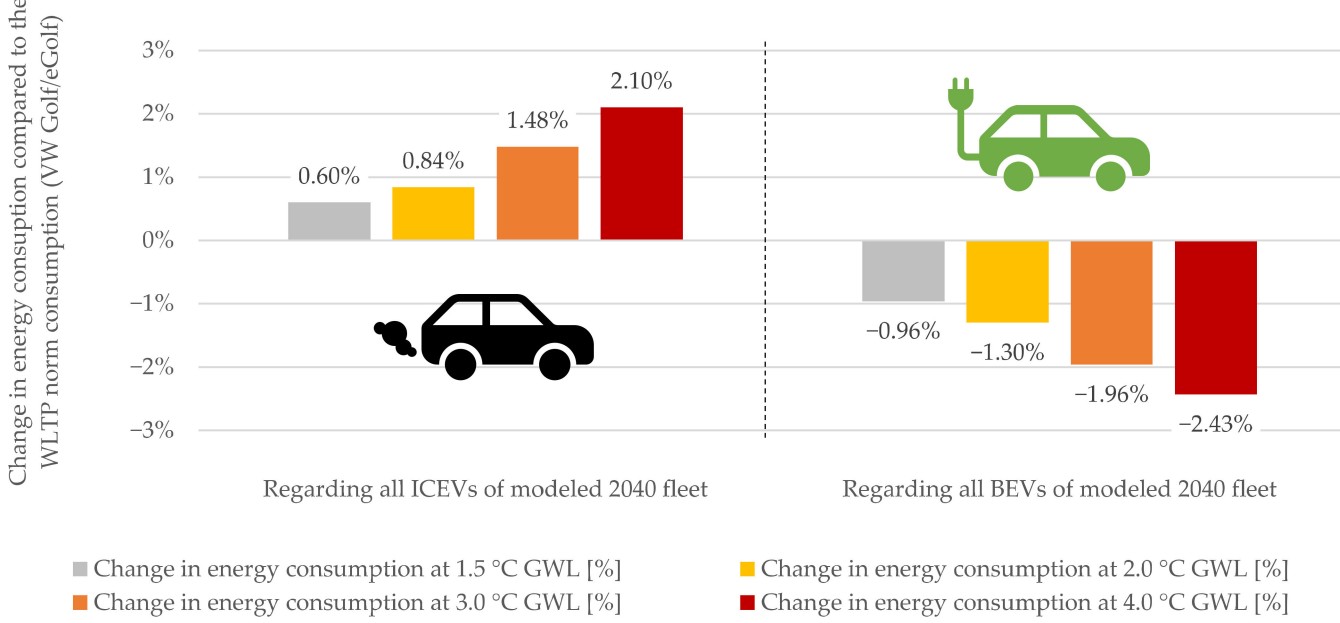

**Figure 6.** Average change in the annual energy consumption of a VW Golf 1.0l eTSI (**left**) and a VW eGolf (**right**) calculated for different global warming levels (GWL) based on the MiD Dataset.

Concerning a country in the central Europe climate zone, like Germany, we can expect a lower overall energy consumption of battery electric vehicles with regard to the climate

change. The BEV benefits from the lower heating needs during winter caused by the global warming. Those energy savings will likely exceed the higher climatization needs during the summer months. At a GWL of +4.0 °C, the calculated energy consumption of a BEV on average is 2.4% lower than without taking the climate-change-related changes in temperature into account.

## 5. Discussion and Limitations

This work deals with the changes in climate on future vehicles. As it is extremely difficult to quantify these various changes, we limited ourselves to the changes relating to vehicle air conditioning and the cold start of the vehicle. We found that the HVAC has a much greater impact on the energy consumption of the vehicle, especially in relation to BEVs. It should be noted, however, that this study is only based on numerical calculations based on average daily temperatures.

Within the scope of the project, it was not possible to verify the results by driving tests in practice. The assumptions regarding the heating and cooling settings are conservative, e.g., solar radiation was not specified as a function of time of day, and the number of people in the vehicle described in the MiD-dataset was also not dynamically considered in the HVAC model. Furthermore, it should be mentioned that a golf model was assumed for each BEV and ICEV in the projected German fleet for 2040 because the Volkswagen Golf is the best-selling passenger car in Germany [33].

No model diversity or additional vehicle concepts, such as plug-in hybrid or fuel-cell-electric vehicles, were considered. Finally, it is important to clarify that 2040 is the reference year of the projected vehicle fleet. In reality, however, different reference dates apply for the different global warming levels. For these model calculations, we assumed that the various GWLs all apply in 2040. This, together with the consideration of more realistic air-conditioning settings in terms of variable solar irradiation and passenger numbers, could be addressed in future studies.

The ambient temperature also affects the density of the air. This decreases with rising temperatures, which in turn decreases the air resistance of the vehicle. However, since this effect applies equally to all vehicles regardless of their drivetrain, it was not considered in the scope of this work. Other effects caused by global warming, such as prolonged periods of heat or heavy rainfall, can also have a significant impact on future vehicles [34]. However, these are extremely difficult to quantify. That is why we limited ourselves to the effect of temperature rise on the energy consumption of vehicles. However, there may, of course, be other effects of climate change on vehicles.

An increase in heat and summer days could reduce the calendar life of electric vehicle batteries [23]. Therefore, new passive heat protection mechanisms could be developed for future vehicles. For example, solar reflective films on vehicle windows [35] or more advanced vehicle paint finishes (without a clear coat, the heating of vehicle sheet metal in the sun can be up to 15 K lower [17]). It is possible to consider additional HVAC modes, such as active ventilation or even climatization when short-term parking, to reduce the heating of the vehicle. Furthermore, parking and especially charging areas should be shaded if possible in order to avoid unnecessary air-conditioning needs in the summer.

## 6. Conclusions

We quantified possible climate change-related changes in the energy consumption of different vehicle drivetrains by focusing on the HVAC and cold start events. Battery electric vehicles will likely benefit from the lower heating needs in winter and show a lower overall energy consumption with higher ambient temperatures in cold and moderate climate zones. Vehicles powered by internal combustion engines, on the other hand, show an increased energy consumption due to the increase in cabin climatization needs, which are not compensated by the lower number of cold start events.

The results demonstrate that, without considering the impact of vehicle manufacturing and recycling, the difference in local emissions caused by ICEV and BEV will increase in

the future due to climate change. As BEVs already in the present climate perform much better than ICEVs in terms of local emissions, this suggests that an accelerated market uptake of BEVs in the upcoming years would be ecologically desirable. The sooner the effects warmer temperatures through climate change come into effect, the higher the environmental benefits of a higher market share of BEVs will be.

**Author Contributions:** S.H.: Conceptualization, Methodology, Data Curation, Software, Visualization, Writing—Original Draft Preparation. A.G.: Data Curation, Writing. S.N.: Data Curation, Visualization. All authors have read and agreed to the published version of the manuscript.

**Funding:** This study was produced within the scope of the Helmholtz Association's mobility subproject for the Climate Initiative.

**Data Availability Statement:** The data of the Mobility in Germany 2017 national household travel survey can be accessed via the Clearing House for Transport Data of the German Aerospace Center against a processing charge (https://www.dlr.de/cs/en/desktopdefault.aspx/tabid-669/1177_read-2160/ (accessed on 20 November 2020)). The weather station data can be retrieved in the open data portal of the German Weather Service (https://www.dwd.de/DE/klimaumwelt/cdc/cdc_node.html (accessed on 20 November 2020)). The data outputs of the regional climate models of the German Climate Service Centre (GERICS) used for this study can generally be accessed via the Euro-Cordex project (https://euro-cordex.net/060378/index.php.en (accessed on 25 November 2020)). However, at the moment of the submission of this paper, the concrete data used have not yet been published.

**Conflicts of Interest:** The authors declare that they have no known competing financial interests or personal relationships that could have appeared to influence the work reported in this paper.

## Appendix A

*Reference Vehicle VW Golf*

The Volkswagen eGolf and the Golf 1.0l eTSI served as reference vehicles. The most important technical data are shown in Table A1.

**Table A1.** Technical data of reference vehicles.

| Parameter | Value | Unit | Source |
|---|---|---|---|
| Energy consumption VW eGolf | 15.8 | kWh/100 km | [36] |
| Fuel consumption VW Golf 1.0l eTSI | 5.2 | l/100 km | [36] |
| Length of WLTP | 23.262 | km | [19] |
| Density gasoline | 745.8 | $kg/m^3$ | [37] |
| Heating value gasoline | 11.75 | kWh/kg | [37] |
| Efficiency of gasoline engine | 0.33 | - | [31] |
| $CO_2$ conversion factor | 2.357 | $kg_{CO_2}/l_{Gasoline}$ | [32] |

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
