# Peer review of "Impact of Climate Change on the Energy Consumption of Passenger Car Vehicles"

_wevj, doi:10.3390/wevj13080146_

Round 1
Reviewer 1 Report
The topic is timely. The authors are recommended to address some main concerns before publication.
Major concerns:
1. Introduction: what is the contribution of this study. The authors are recommended to highlight the innovations and contributions of this paper to the practice and the literature.
2. A comprehensive discussion on related literature should be included to demonstrate that the research topic is meaningful enough and your methodology used is appropriate.
3. Figure 4, only three starting temperature points for each type of vehicle are calculated (as dots and triangles). Is it enough to determine the function of trendline?
4. This study investigates the impact of ambient temperature on the energy consumption of automobiles based on only numerical calculations, and the results are not supported by actual cycle experiments. Therefore, I am afraid that the results of the study are not convincing enough.
5. The energy consumption of cabin climatization must varies with the type of vehicle and the powers of air conditioners and heaters. There are so many brands of passenger cars around the world. Without control of these variables, how do the authors determine the amount of energy consumption induced by climate warming?
Minor concerns:
1. Figure 1 is very blurry and makes it difficult to identify the green grid.
2. Line 160, the authors should clarify the settings of passenger number and sun flux during heating and cooling while calculating cabin thermal energy needs. Four passengers are considered during cooling but are not taken into consideration during heating. Is it reasonable of assuming a constant solar irradiation?
Author Response
First of all, we would like to thank the reviewers for their efforts. We found their comments quite helpful and implemented most of the required changes in different ways as outlined in the attached document. We believe that this substantially improved the quality of our paper and are looking forward to seeing the reactions on the resubmitted version.

Reviewer 2 Report
Line 29: you may mention, that also the different air density at different ambient conditions has a significant impact but this is not considered in your analysis. The effect is ca 1% lower air resistance at 3°C temperature increase.
Line 105: for HVAC not just average temperatures are relevant but the temperature distribution. For average German weather, i.e. 12°C, one needs almost no heating and no cooling. Furthermore the humidity is very important for AC power demand due to the high heat capacity of water and even more the sun radiation is relevant for cooling or heating power demand. These parameters should at least be mentioned and the settings of these parameters should be clear (see later comments on sun radiation).
In this chapter 2.1 it is unclear, what the result of the extensive data analysis is. Is it a matrix of temperatures and corresponding number of starts and trip length distribution? Please explain and/or show a graph or a table.
Line 159: please describe, what you mean with “transient power demand”. Is this initial cool down or heating demand after vehicle start when soaking at low or high temperatures?
Line 161: the 700 W/m² sun radiation is representative for Germany only for time slots between ca 11:00 to 15:00 in summer months, according to my knowledge. You should state somewhere the shares of driving time under this condition. Also 4 passengers are not average values. Have you made an assessment of the impact? More passengers would reduce the heating demand but increase cooling power.
Figure 2: please define “thermal power”: is this the difference of Enthalpy from fresh air to air conditions at the inlet vents or is this mechanical power for AC compressor? If so, is it including the fan for ventilation and which air mass flows are assumed?
I assume your results refer to some automatic HVAC settings, since results for fresh air intake and recirculation of cabin air make huge differences in power demands. Do you know the settings your curves refer to? If so, please specify otherwise mention this uncertainty.
Line 222: please add information on the test cycles or at least on the cycle length for Zhu et.al. You should consider the cycle length when comparing cold start extra emissions. The longer the trip, the lower the cold start extra consumption is in %. Using the extra consumption (g per start or kWh/start) is a better unit. This is also relevant for your figure 4. These values obviously refer to WLTP and thus to trips with ca. 23km. I suggest you amend the legend for figure 4 to “Percentage increase in fuel consumption caused by cold starts at different temperatures in the WLTP compared to a norm start at 22°C…..”. You deal with cycle length impacts in chapter 3.2. Maybe you refer to this chapter.
For the DLR tests: Have the tested vehicles been equipped with an electric heater for fast cabin heating? If yes, is the additional energy consumption considered in the results, i.e. was the SOC difference of the battery before and after test evaluated?
Line 317: please specify the fan power also for temperatures above 20°C. This should somehow increase with increasing temperature.
Figure 5: “colt start” instead of “cold start” in both graphs
Chapter 6: you should mention, that a constant 700 W/m² sun radiation is also a simplification. Question: did you use the 700W/m² value for the entire year? If so, heating demand would be underestimated and cooling overestimated. A quick estimation for normal glazing suggests for 20°C for 100W sun radiation ca 150W HVAC demand while it is 360 W for 700W sun radiation.
Author Response

(The authors gave the same response as above.)

Reviewer 3 Report
The paper proposes an analysis of the impact on energy consumption of the climate changes, using data obtained in a large study involving the German population. The influence of different delta temperatures is studied both on a conventional and an electric vehicles.
The reviewer would positively comment on the methodology explained in this paper, however minor revisions are recommended to improve the quality of the study.
A more detailed transcript of other possible changes/suggestions is hereafter reported for the authors’ convenience.
· As a general comment, the reviewer would recommend to improve the literature review to exhaustively understand the contributions of this paper, that are not completely clear now.
· It would be helpful to add a final section in the introduction where to add what it is found in the next chapters and guide the reader through the paper.
· The reviewer would not recommend using words as “very high” or “very likely”, they might not be formal. Some possible changes could include “extremely” or “excessive” for very high. Please modify the text accordingly.
· In figure 4, the legend should say “Fontaras et al. [2]”, not [4] as the reference is wrong.
· In the reviewer’s opinion, “accelerated market uptake of BEVs […] would be ecologically desirable” is a strong opinion that does not take into account a large variety of factor. It is true that tailpipe emissions are null, however one should look at a larger picture and include the life cycle assessment. This means, as an example, considering related CO2 emissions for recycling and manufacturing a battery. Please modify what stated in the conclusions to include this “bigger picture”.
Author Response

(The authors gave the same response as above.)

Round 2
Reviewer 1 Report
I am satisfied with the detailed responses and edits made by the authors.